# Cross-Domain Knowledge Transfer for Sustainable Heterogeneous Industrial Internet-of-Things Networks

**DOI:** 10.3390/s24113265

**Published:** 2024-05-21

**Authors:** Zhenzhen Gong, Qimei Cui, Wei Ni

**Affiliations:** 1National Engineering Laboratory for Mobile Network Technologies, Beijing University of Posts and Telecommunications, Beijing 100876, China; gongzhenzhen0822@gmail.com; 2Data61, Commonwealth Science and Industrial Research Organization (CSIRO), Marsfield, NSW 2122, Australia; wei.ni@data61.csiro.au

**Keywords:** industrial internet-of-things (IIoT), age of information (AoI), energy efficiency, cross-domain

## Abstract

In this article, a novel cross-domain knowledge transfer method is implemented to optimize the tradeoff between energy consumption and information freshness for all pieces of equipment powered by heterogeneous energy sources within smart factory. Three distinct groups of use cases are considered, each utilizing a different energy source: grid power, green energy source, and mixed energy sources. Differing from mainstream algorithms that require consistency among groups, the proposed method enables knowledge transfer even across varying state and/or action spaces. With the advantage of multiple layers of knowledge extraction, a lightweight knowledge transfer is achieved without the need for neural networks. This facilitates broader applications in self-sustainable wireless networks. Simulation results reveal a notable improvement in the ’warm start’ policy for each equipment, manifesting as a 51.32% increase in initial reward compared to a random policy approach.

## 1. Introduction

The sustainability of communication networks is a critical goal for next-generation wireless systems (e.g., 6G and beyond [1]). Network sustainability is defined as an approach that successfully integrates and balances environmental responsibility, economic viability, and social equity. Despite the growing attention and hype surrounding the sustainability of 6G, there is a lack of a rigorous and practical definition to guide its implementation in networks. Sustainability has been mainly linked to green networking to achieve the United Nations’ Sustainable Development Goals (SDGs) [2]. In practice, this is particularly related to energy efficiency of the versatile network elements. In particular, smart factories constitute a significant component in Industrial Internet-of-Things (IIoT) [3] and Industry 4.0 [4], playing a key role in enabling cyber-physical systems to function autonomously. IIoT applications typically requires the automation of a large number of devices in manufacturing with limited hardware capabilities and energy resources, usually with small batteries [5]. Industrial 4.0 [6] encompasses emerging technologies, such as artificial intelligence (AI), edge computing, and digital twin (DT) and so on. In particular, the work in [7] comprehensively investigated the intelligence maintenance in various aspects of maintenance. Specifically, it focused on the human-in-the-loop-based maintenance and its role in enhancing physical resilience in smart manufacturing. This paradigm requires increased flexibility, agility and resilience through the lifespan of the IIoT devices. Consequently, in the realm of IIoT, the smart factories are expected to integrate advanced autonomous capabilities along with enhanced energy-efficient functionality. Nevertheless, the robots, sensors and actuators in the factories are empowered with different sources of energy. Such sources include power grids [8], renewable technologies [9] (e.g., solar), and other energy harvesting techniques [10] (e.g., radio frequency (RF) energy). Subsequently, ensuring the energy efficiency of each individual equipment necessitates adopting a unique mode of operation that is specifically tailored to the varying availability and abundance of their respective energy sources. This can have a direct implication on other critical performance metrics of operation in smart factory. Chief among these metrics is the *age of information (AoI)* [11] that represents the degree of freshness of the data acquired from the monitored autonomous physical systems [12]. With a focus on both energy efficiency and information freshness, the sustainability of each individual equipment can be significantly enhanced. However, assuring the sustainability of the IIoT as a whole requires looking beyond the individual equipment. In fact, the overall performance and environmental impact of the IIoT will crucially depend not only on the performance of single piece of equipment but also on long-term environmental friendliness of its solution. This encompasses considerations of the system’s overall energy consumption and its ability to sustain prolonged operation without causing harmful impacts on the environment, by considering the associated complexity and energy efficiency of the solution.

The minimization of hybrid energy sources in smart factories has been extensively investigated in various scenarios [13,14]. For instance, the works in [13,14] study the minimization of grid energy consumption in a mixed energy supply scenario. Nonetheless, these works leverages reinforcement learning (RL) solutions [15] that assume a homogeneous model across equipment having heterogeneous energy utilities. In fact, these studies often assume uniformity of state and/or action spaces between heterogenous scenarios, which can barely hold true with the unique operation associated to each equipment [16]. Therefore, in practical real-world scenarios, a robust RL approach is needed to effectively address the heterogeneous nature of the cyber-physical system, while ensuring the sustainability of the solution. Notably, one should consider an RL solution that generalizes across multiple tasks. For instance, the works in [17,18] employ multiple experts to optimize the aggregated performance across different groups. However, the use of multiple agents hinders knowledge sharing among these groups and leads to increased costs as the number of groups grows. The work in [19] considers a federated imitation learning method for cross-domain knowledge sharing framewor. However, the utilization of neural networks slows down the learning process. Furthermore, the application of gradient descent (GD) [20] in such operations incurs additional energy costs as it requires a significant amount of resources to converge. Consequently, to ensure network sustainability, encompassing both the energy efficiency of individual equipment and the computational efficiency of the entire network, a more universally applicable and generalizable solution is essential for heterogeneous Internet of Things (IoT).

The main contributions of this paper is the development of a globally generalizable RL solution, designed to enhance the overall sustainability of cyber-physical systems comprising heterogeneous energy sources. In particular, we tackle the sustaibality issues at both the equipment and system levels by introducing a lightweight, cross-domain knowledge sharing solution. This innovative approach leverages a three-layered knowledge repository structure to facilitate efficient knowledge storage and transfer across the system. Numerical simulations demonstrate that the proposed method consistently outperforms other baseline methods in computational complexity while maintain a comparable performance for smart factories.

The rest of this paper is organized as follows. The system models and problem formulation are provided in Section 2. The proposed cross-domain knowledge sharing framework and the corresponding solutions are presented in Section 3. Simulation results are given in Section 4. Finally, conclusions and future works are drawn in Section 5.

## 2. System Models

Consider a set N of *N* smart factory equipment having heterogeneous energy resources in a smart factory. These pieces of equipment are distributed for various manufacturing purposes such as supply chain integration, pre-production setup, production, quality control and inspection, packaging and storage, delivery and so on. Each equipment collects sensory data from its surrounding environment and subsequently executes actions that are tailored to the information gathered. As illustrated in Figure 1, these pieces of equipment are clustered into three distinct groups according to their energy sources. We use x∈X to index the three groups such that x=1,2,…,X, whereby each group includes a set $N_{x}$ of $N_{x}$ equipment. In particular, three sources of energy supply are considered: (i) *grid power (GP), (ii) green sources (GS), and (iii) mixed sources (MS)*. Specifically, MSs encompasses both the grid and harvested energy resources. In addition, cyber-physical equipment within each group collects data packets from their respective surrounding environments and abstract useful information using their processing capabilities. As illustrated in Figure 2, the abstracted information is subsequently transmitted to a nearby base station (BS).

We consider a time-slotted system where each timeslot has a uniform length denoted as τ. These timeslots are indexed sequentially as t=1,2,…,T. A Rayleigh fading channel is considered for the uplink communication between smart factory equipment and BS. The data transmission rate $\phi (c_{y(t)})$ for each equipment y∈N at time slot *t* can be obtained as below: (1)$$\phi \left(c_{y}\left(t\right)\right)=B\log_{2}\left(1+\frac{gp_{y}\left(t\right)}{I+BN_{0}}\right),\;\forall y\in N,$$
where $c_{y}\left(t\right)$ is the number of bits to be processed, $\phi (c_{y}\left(t\right))$ is the number of bits to be transmitted after processing, $p_{y}\left(t\right)\in \left[0,p_{y,\max}\right]$ (in dBm) is the transmitter power used to upload the abstracted information, *B* is the channel bandwidth, *I* is the interference from other pieces of equipment in corresponding group, $N_{0}$ is noise power spectral density, *g* is the channel response, which is related to the distance between each equipment *y* and the BS, i.e., $l_{y}$. Next, we present the energy models of each group based on their energy sources:*GP Source*: GP typically refers to power that is supplied through an electrical grid. Hence, GP-powered equipment does not have energy limitations. For instance, the robots and actuators in production line are connected to grid energy supply. The energy consumption $e_{i}\left(t\right)$ of each equipment $i\in N_{1}$ can be divided into two categories: (a) transmission energy $e_{i}^{T}\left(t\right)=\tau p_{i}\left(t\right)$ consumed to transmit abstracted information to the BS and (b) computing energy $e_{i}^{C}\left(t\right)=\varsigma \kappa_{i}\vartheta^{2}c_{i,t}$ used to process the collected data packets:
(2)$$e_{i}\left(t\right)=e_{i}^{T}\left(t\right)+e_{i}^{C}\left(t\right)=\tau p_{i}\left(t\right)+\varsigma \kappa_{i}\vartheta^{2}c_{i,t},$$
where ς is the energy consumption coefficient depending on the chip of each IIoT equipment, $\kappa_{i}$ is the number of central processing unit (CPU) cycles required for processing per bit data, assumed to be equal for all pieces of equipment and ϑ is the frequency of the CPU clock of each equipment [21].*GS Source*: Renewable energy sources, such as wind power, solar power, thermal power and RF are used to enable the establishment of a self-sustainable green network. For example, drones and robots utilized for quality inspection and automated delivery systems are predominantly powered by battery technology. This reduces dependence on conventional grid energy and, consequently, enhances the mobilities while offering greater flexibility and efficiency in operational processes. These energy harvesting methods consistently capture energy from natural environments, converting it into electrical power and storing the collected energy in rechargeable batteries. We define $E_{\max}$ as the maximum amount of energy that can be stored in a battery. When the battery reaches its full capacity, any additional harvested energy will be discarded. Consider an ideal rechargeable battery with no energy loss during storage or retrieval processes. At each time slot, the harvested energy $e_{j}^{h}\left(t\right)\geq 0$ by equipment $j\in N_{2}$ follows follows a Bernoulli distribution with probability σ∈[0,1], such that:
(3)$$e_{j}^{h}\left(t\right)=p^{\mathrm{solar}}\times \epsilon_{0}\times \epsilon_{1}\times \epsilon_{2}\;,$$
where $p^{\mathrm{solar}}$ is the density of solar power to the equipment [22]. We consider a typical solar-powered equipment that is equipped with a photovoltaic panel with size $\epsilon_{0}$ and the energy transfer efficiency $\epsilon_{1}$. Considering the heterogeneity in solar power density, a uniformly distributed random variable, $\epsilon_{2}$ is taken into account. Consequently, the energy level of the battery $e_{j}^{b}\left(t\right)$ will be given by:
(4)$$e_{j}^{b}\left(t+1\right)=\min\left\{E_{\max},\;e_{j}^{b}\left(t\right)+e_{j}^{h}\left(t\right)-e_{j}^{T}\left(t\right)-e_{j}^{C}\left(t\right)\right\},$$
where $e_{j}^{T}\left(t\right)$ and $e_{j}^{C}\left(t\right)$ are transmission energy and computing energy for equipment *j*. Moreover, the following constraint stands:
(5a)$$\begin{array}{cc} 0\leq e_{j}^{b}\left(t\right) & \leq E_{\max} \end{array}$$
(5b)$$\begin{array}{cc} e_{j}^{T}\left(t\right)+e_{j}^{C}\left(t\right)= & e_{j}\left(t\right)\leq e_{j}^{b}\left(t\right), \end{array}$$
where $e_{j}\left(t\right)$ is the energy consumption at time slot *t* for equipment in group 2. ([Disp-formula FD5a-sensors-24-03265]) implies the battery limitation of equipment $j\in N_{2}$. ([Disp-formula FD5b-sensors-24-03265]) implies that the available energy, which can be used for processing and transmitting energy at the beginning of each time slot, must not exceed the energy level of the battery.*MS Source*: The third group of cyber-physical equipment is powered by hybrid energy sources comprising both the grid and renewable energy sources. For example, the industrial sensors are strategically deployed to monitor a range of environmental parameters as well as the status of products. This design aims to reduce energy consumption from the grid power while mitigating the randomness and intermittency associated with green energy. Accordingly, for an equipment $k\in N_{3}$, the consumed energy at time slot *t* comprises two sources: grid energy $e_{k}^{G}\left(t\right)$ and battery energy $e_{k}^{B}\left(t\right)$. We assume the same energy harvesting model as previously defined, such that $e_{k}^{h}\left(t\right)$ is updated as in (Equation 3). Different from GS, the battery level is updated as:
(6)$$e_{k}^{b}\left(t+1\right)=\min\left\{E_{\max},\;e_{k}^{b}\left(t\right)+e_{k}^{h}\left(t\right)-e_{k}^{B}\left(t\right)\right\}$$
where $e_{k}^{b}\left(t\right)$ is the battery level at each time slot. Furthermore, the following constraints are held:
(7a)$$\begin{array}{cc} e_{k}^{T}\left(t\right)+e_{k}^{C}\left(t\right) & \leq e_{k}^{B}\left(t\right)+e_{k}^{G}\left(t\right)\;, \end{array}$$
(7b)$$\begin{array}{cc} e_{k}^{B}\left(t\right) & \leq e_{k}^{b}\left(t\right)\;, \end{array}$$
(7c)$$\begin{array}{cc} e_{k}\left(t\right) & =e_{k}^{G}\left(t\right) \end{array}$$
(7d)$$\begin{array}{cc} 0\leq e_{k}^{b} & \left(t\right)\leq E_{\max} \end{array}$$
where $e_{k}^{T}\left(t\right)$ and $e_{k}^{C}\left(t\right)$ are the transmission and computing energy separately. Moreover, ([Disp-formula FD7a-sensors-24-03265]) implies that the consumed energy must not exceed the total energy provided by both the battery and the grid. ([Disp-formula FD7b-sensors-24-03265]) implies that the permissible battery energy must not exceed the available battery capacity. Since our optimization objective is to minimize grid energy consumption, we set the energy optimization variable $e_{k}\left(t\right)$ equal to $e_{k}^{G}\left(t\right)$ in ([Disp-formula FD7c-sensors-24-03265]). ([Disp-formula FD7d-sensors-24-03265]) indicates the battery limitation of each equipment $k\in N_{3}$.

### AoI Model for Heterogeneous Scenarios

At each time slot, sensing data packets arrive at the equipment with a probability $\lambda_{y}$. The size of the data packet, denoted as $a_{y}\left(t\right)$, follows a Gaussian distribution. Data packets, once collected, are placed in a waiting queue. The system processes and transmits these packets employing a first-come-first-served (FCFS) approach. Consequently, the queue size $b_{y}\left(t\right)$ can be updated as follows: (8)$$b_{y}\left(t+1\right)=b_{y}\left(t\right)+a_{y}\left(t\right)-c_{y}\left(t\right).$$

The AoI at time slot *t* is defined as the timestamp of the most recently processed and successfully received packet at the receiver. The entire process encompasses both data processing time and transmission time. Formally, the update of AoI $\Delta_{y}\left(t\right)$ is as follows: (9)$$\Delta_{y}\left(t+1\right)=\left\{\begin{array}{cc} \Delta_{y}\left(t\right)+1\;, & \mathrm{if}\;\varpi_{y}\left(t\right)=1\\ \min\{\left(t+1\right)-U\left(t\right),\Delta_{\max}\}\;, & \mathrm{otherwise}. \end{array}\right.$$
where, U(t) represents the generation timestamp of the most recent packet and $\Delta_{\max}$ is the maximum AoI value and $\varpi_{y}\left(t\right)=1$ indicates that the processing of a packet is finished. This limit is imposed to constrain the impact of AoI on performance after a certain level of staleness is reached.

With the aforementioned models, we can now proceed to our optimization objective. For each equipment, we define a cost function $w_{y}\left(t\right)=\eta_{1}\Delta_{y}\left(t\right)+\left(1-\eta_{1}\right)e_{y}\left(t\right)$, where $\eta_{1}$ is a tradeoff factor to balance AoI and energy cost and $e_{y}\left(t\right)$ represents the energy cost of any equipment of three groups. Our objective is to minimize the averaged cost $w_{y}\left(t\right)$ of all pieces of equipment throughout all the time, which can be written as:
(10)$$\begin{array}{c} \min_{\left\{p_{y}\left(t\right)\right\},\left\{e_{y}^{G}\left(t\right)\right\},\left\{e_{y}^{B}\left(t\right)\right\}}\;\sum_{x=1}^{3}\frac{1}{N_{x}}\sum_{y\in N_{x}}w_{y}\left(t\right) \end{array}$$
(10a)$$\begin{array}{c} s.t.\;0\leq p_{y}\left(t\right)\leq p_{y,\max} \end{array}$$
(10b)$$\begin{array}{c} 0\leq \Delta_{y}\left(t\right)\leq \Delta_{\max} \end{array}$$
(10c)$$\begin{array}{c} \left(2\right),\;\forall y\in N_{1} \end{array}$$
(10d)$$\begin{array}{c} \left(3\right),\;\left(4\right),\;\left(5\right),\;\forall y\in N_{2} \end{array}$$
(10e)$$\begin{array}{c} \left(6\right),\;\left(7\right),\;\forall y\in N_{3}\;, \end{array}$$
where ([Disp-formula FD10a-sensors-24-03265]) indicates that the transmission power must not surpass the maximum power of each equipment, ([Disp-formula FD10b-sensors-24-03265]) indicates the limitation requirements of AoI. ([Disp-formula FD10c-sensors-24-03265]) implies the conditions of equipment $i\in N_{1}$ while ([Disp-formula FD10d-sensors-24-03265]) and ([Disp-formula FD10e-sensors-24-03265]) are constraints for all pieces of equipment in group GS and MS, separately.

## 3. Cross-Domain for Heterogeneous Scenarios

Problem (Equation 10) is NP complete, making it inherently computationally expensive. Furthermore, the unique constraints specified in ([Disp-formula FD10c-sensors-24-03265]), ([Disp-formula FD10d-sensors-24-03265]) and ([Disp-formula FD10e-sensors-24-03265]) add to the complexity of this problem. Additionally, this problem is compounded by the absence of any presupposed knowledge regarding the distribution patterns of data and energy arrivals. To tackle these challenges, we adopt RL, a method that does not require prior knowledge of the underlying distribution patterns. First, three distinct RL models are presented for each group. Then, the overall minimization problem in (Equation 10) is considered as a cross-domain knowledge sharing problem.

### 3.1. Markov Decision Processes (MDPs) Models

Initially, the manufacturing related equipment is partitioned into three distinct groups. As such, the objective of each group is to minimize the averaged cost for all pieces of equipment. Without interfering the overall objective in (Equation 10), we model the objectives of each group using MDPs:*GP Source*: The MDP tuple of the first group can be presented as $\left(S_{i},A_{i},R_{i}\right)$, where $S_{i}$ is the state space and $A_{i}$ is the action space and $R_{i}$ is the reward function separately. Particularly, $S_{i}=\left\{s_{i,t}\right\}=\left\{\Delta_{i,t},b_{i,t}|\Delta_{i,t}\in \left[0,\Delta_{\max}\right],\;b_{i,t}\in N\right\}$. The action space is the set of all possible transmitting powers such that $A_{i}=\left\{a_{i,t}\right\}=\left\{p_{i,t}|p_{i,t}\in \left[0,p_{i,\max}\right]\right\}$. The reward function can be defined as $r_{i}\left(s_{i,t},a_{i,t}\right)=w_{i}\left(t\right)$. The parameterized policies can be defined as $\pi_{\theta_{i}}\left(a_{i,t}|s_{i,t}\right)=\mathrm{Pr}\left\{a_{i,t}|s_{i,t},\theta_{i}\right\}$, where $\theta_{i}\in R^{d_{1}}$, with $d_{1}=2$.*GS Source*: The MDP tuple of the second group can be presented as $\left(S_{j},A_{j},R_{j}\right)$. Particularly, $S_{j}=\left\{s_{j,t}\right\}=\left\{\Delta_{j,t},b_{j,t},e_{j,t}^{b}|\Delta_{j,t}\in \left[0,\Delta_{\max}\right],\;b_{j,t}\in N,\;e_{j,t}^{b}\in \left\{0,E_{\max}\right\}\right\}$. The action space is the the same as group 1, such that $A_{j}=\left\{a_{j,t}\right\}=\left\{p_{j,t}|p_{j,t}\in \left[0,p_{j,\max}\right]\right\}$. Similarly, the reward function and the parameterized policies can be defined as $r_{j}\left(s_{j,t},a_{j,t}\right)=w_{j}\left(t\right)$ and $\pi_{\theta_{j}}\left(a_{j,t}|s_{j,t}\right)=\mathrm{Pr}\left\{a_{j,t}|s_{j,t},\theta_{j}\right\}$, where $\theta_{j}\in R^{d_{2}}$, with $d_{2}=3$.*MS Source*: The MDP tuple of the third group can be presented as $\left(S_{k},A_{k},R_{k}\right)$. Particularly, $S_{k}=\left\{s_{k,t}\right\}=\left\{\Delta_{k,t},b_{k,t},e_{k,t}^{b}|\Delta_{k,t}\in \left[0,\Delta_{\max}\right],\;b_{k,t}\in N,\;e_{k,t}^{b}\in \left\{0,E_{\max}\right\}\right\}$. The action space is $A_{k}=\left\{a_{k,t}\right\}=\left\{p_{k,t},e_{k,t}^{B},e_{k,t}^{G}|p_{j,t}\in \left[0,p_{j,\max}\right],e_{k,t}^{B}\in \left[0,E_{\max}\right],e_{k,t}^{G}\in N\right\}$. With the similar reward function $r_{k}\left(s_{k,t},a_{k,t}\right)=w_{k}\left(t\right)$, the parameterized policies can be defined as $\pi_{\theta_{k}}\left(a_{k,t}|s_{k,t}\right)=\mathrm{Pr}\left\{a_{k,t}|s_{k,t},\theta_{j}\right\}$, where $\theta_{k}\in R^{d_{3}}$, with $d_{3}=9$.

So far, we have successfully formulated the cost tradeoff between energy consumption and AoI as a series of MDPs, each corresponding to an individual equipment. For each equipment *y*, the optimization target can be writen as $\min_{\theta_{y}}J\left(\theta_{y}\right)=E\left[1/T\sum_{t=1}^{T}r_{y,t}\left(s_{y,t},a_{y,t}\right)\right]$. While RL demonstrates the capability to learn and optimize for each equipment individually, the scalability of this approach becomes a concern in large-scale factories due to the potentially large number of learning agents involved. More importantly, this approach can be both time-consuming and energy-intensive. Consequently, there is a pressing need for a more efficient method that can collectively optimize across all pieces of equipment in the three distinct groups. Such a method is crucial not only for the energy efficiency of these equipment but also for the overall sustainability of the cyber-physical system.

### 3.2. Cross-Domain Knowledge Sharing

To facilitate learning and knowledge sharing across multiple groups, techniques like multi-task learning (MTL) and meta-learning are employed. These methods are adept at managing the simultaneous learning of multiple tasks. However, a significant limitation of these methods is their inherent assumption of model consistency across groups. This assumption poses a challenge when optimizing groups across heterogeneous groups, especially when there is a variation in the state and action spaces of these groups. Consequently, given the substantial resource costs associated with the requirement for numerous learning agents, the need for an efficient cross-domain knowledge transfer method becomes increasingly apparent.

To facilitate knowledge sharing among groups and within each group, a three-layered knowledge base is designed as in Figure 3. A global knowledge base, $L\in R^{d\times m}$, serves as the shared knowledge among groups. Three group-based knowledge matrices, denoted as $G_{x}\in R^{d_{x}\times d}$, where x∈{1,2,3}, are also utilized to store the knowledge specific to each group. These matrices serve as a bridge between the global knowledge base and equipment-specific mapping vectors, represented by $s_{y}\in R^{m}$, where y∈N. As demonstrated by the MDP models of each group, the state and action spaces of each group can vary. In other words, the dimensions of the MDP policies, i.e., $\theta_{i}$, $\theta_{j}$, and $\theta_{k}$, have different dimensions $d_{x}$, as illustrated in Figure 3. However, due to the existence of varied group knowledge bases $G_{x}$, the variations of the policy vectors are mapped to the same space. This enables the achievement of a global knowledge base L that can be shared across different domains. As a result, the policy parameters of each equipment can be obtained as: (11)$$\theta_{y}=G_{x}*L*s_{y}\;.$$

Accordingly, our objective in (Equation 10) with the three-layered knowledge system can be represented as the minimization problem: $$g\left(L,G_{x}\right)=\sum_{x=1}^{X}\left\{\frac{1}{N_{x}}\sum_{y\in N_{x}}\min_{s_{y}}\left[J\left(\theta_{y}\right)+\mu_{1}{∥s_{y}∥}_{1}\right]+\mu_{2}{∥G_{x}∥}_{F}^{2}\right\}+\mu_{3}{∥L∥}_{F}^{2}$$
where $L_{1}$-norm approximates the vector sparsity and ${∥L∥}_{F}={\left(tr\left(LL^{\prime }\right)\right)}^{1/2}$ is the Frobenius norm of matrix L. The parameter $\mu_{1}$ controls the balance between the policy’s fit and the feature’s fit. Also, $\mu_{2}$ and $\mu_{3}$ are two regularization parameters, where $\mu_{2}$ controls the sparsity of $s_{y}$. The penalty on the Frobenius norm of G and L regularizes the predictor weights to have low $L_{2}$-norm and avoids overfitting.

The above objective can be approximated by performing a second-order Taylor expansion towards $J(\theta_{y})$ around the optimla policy $\alpha_{y}$, which can be obtained using regular RL methods, such as policy gradient: $\alpha_{y}=\arg\min_{\theta_{y}}J\left(\theta_{y}\right)$. By operating first derivative and second derivative to $J(\theta_{y})$, the above equation can be rewriten as:$$\hat{g}\left(L,G_{x}\right)=\sum_{x=1}^{X}\left\{\frac{1}{N_{x}}\sum_{y\in N_{x}}\min_{s_{y}}\left[∥\alpha_{y}-G_{x}Ls_{y}{∥}_{\Gamma_{y}}^{2}+\mu_{1}{∥s_{y}∥}_{1}\right]+\mu_{2}{∥G_{x}∥}_{F}^{2}\right\}+\mu_{3}{∥L∥}_{F}^{2}$$
where $\Gamma_{y}$ is the Hessian matrix and $∥\alpha_{y}-G_{x}Ls_{y}{∥}_{\Gamma_{y}}^{2}={\left(\alpha_{y}-G_{x}Ls_{y}\right)}^{⊤}\Gamma_{y}\left(\alpha_{y}-G_{x}Ls_{y}\right)$. The constant term was ignored because it has no effect on the minimization. The linear term was ignored because the $\alpha_{y}$ is the estimated optimal policy.

In further, we can split the above equation by all the equipment. Such that, we only optimize the equipment specific $\theta_{y}$ while fix the value of $\theta_{y}$ for all other equipment. The improvement of L and $G_{x}$ can be reflected to other equipment. As such, we can obtain the update function of L and $G_{x}$: (12)$$\Delta L\left(k\right)=\beta_{1}\left[\sum_{x=1}^{3}\frac{1}{|Z_{x}\left(k\right)|}\sum_{z\in Z_{x}\left(k\right)}\left(-G_{x}^{⊤}\Gamma_{z}\alpha_{z}s_{z}+G_{x}^{⊤}\Gamma_{z}G_{x}Ls_{z}s_{z}^{⊤}\right)+\mu_{3}L\right]\;,$$
(13)$$\Delta G_{x}\left(k\right)=\beta_{x}\left[\frac{1}{|Z_{x}\left(k\right)|}\sum_{z\in Z_{x}\left(k\right)}\left(-\Gamma_{z}\alpha_{z}{\left(Ls_{z}\right)}^{⊤}+\Gamma_{z}G_{x}\left(Ls_{z}\right){\left(Ls_{z}\right)}^{⊤}+\mu_{2}G_{x}\right)\right]\;,$$
where, $\beta_{1}$ and $\beta_{x}$, ∀x∈{1,2,3}, are the learning rates for L and $G_{x}$, separately. $z\in Z_{x}$ is the set of observed equipment for each group. Such that, L(k+1)=L(k)+ΔL(k) and $G_{x}\left(k+1\right)=G_{x}\left(k\right)+\Delta G_{x}\left(k\right)$, where *k* means *k*-th update step. With the updated global knowledge base and group base, $s_{y}$ can be obtained by solving a Lasso: (14)$$s_{y}\left(k+1\right)\leftarrow \arg\min_{s_{y}\left(k\right)}\ell \left(G_{x}L,s_{y}\left(k\right),\alpha_{y},\Gamma_{y}\right)\;,$$
where $\ell \left(G_{x}L,s_{y},\alpha_{y},\Gamma_{y}\right)=∥\alpha_{y}-G_{x}Ls_{y}{∥}_{\Gamma_{y}}^{2}+\mu_{1}{∥s_{y}∥}_{1}$. Consequently, the full algorithm can be organized as in Algorithm 1: (1) Initialize the L, $G_{x}$ and $\alpha_{y}$ for all equipment. (2) Estimate $\alpha_{y}$ for all equipment. (3) Randomly choose a piece of equipment *y* and update L and corresponding $G_{x}$ using (Equation 12) and (Equation 13). (4) Compute $s_{y}$ according to (Equation 14). (5) Repeat steps (3) and (4) until the time period comes to an end. It is worth noting that, at each step, we update only the global knowledge base and the corresponding group knowledge base. The performance improvement of equipment in other groups can benefit from the updating of the global knowledge base. In this case, we have $N=Z_{1}\cup Z_{2}\cup Z_{3}$ and $\theta_{y}=G_{x}Ls_{y}$.

**Algorithm 1** Overview of the Proposed Algorithm**Require:** T←0, $L\leftarrow {\mathrm{zeros}}_{d,m}$, $G_{x}\leftarrow {\mathrm{zeros}}_{d_{x},d}$, ∀x∈{1,2,3}**Require:** $\alpha_{y}$, $s_{y}$ for all pieces of equipment    **while** t≤T **do**        Randomly choose a piece of equipment        Identify the group of the chosen equipment as x∈{1,2,3}        Obtain interaction history and compute $\Gamma_{y}$        Update L, $G_{x}$ using (Equation 12) and (Equation 13)        Update $s_{y}$ for device *i* using (Equation 14)        t←t+1    **end while**

### 3.3. Computing Complexity

Each update begins with the computing of $\theta_{y}$ and $\Gamma_{y}$ for each individual equipment. We adopt a base-learner, specifically the episodic Natural Actor Critic (eNAC), characterized by a computational complexity of $O(\xi \left(d_{x},n_{t}\right))$ for each step. Here, $n_{t}$ represents the number of trajectories obtained for a piece of equipment during the current iteration. The update of L includes multiplication of matrix and vectors, which yields $O(d_{x}^{3}+d_{x}dm+m^{2})$ for each step. Similarly, the update of $G_{x}$ has a complexity $O(d_{x}dm+m^{2}+d_{x}^{2})$. The update of $s_{y}$ requires solving an instance of Lasso, which typically would be $O(d^{3}h^{2}+md_{x}^{2}+d_{x}m^{2})$. Therefore, the overall complexity of each update for an individual equipment is $O(d_{x}^{3}+d_{x}dm+md_{x}^{2}+d_{x}m^{2}+\xi \left(d_{x},n_{t}\right))$.

## 4. Simulation Results

### 4.1. Simulation Settings

For our simulations, we consider a circular network area with a radius of 500m and one BS at its center serving three groups of equipment. Each group is distributed in a circle with a radius of 250m. Within each group, we consider $N_{x}=10$ pieces of equipment uniformly distributed. For each group, we have the following simulation parameters:GP: This group of equipment relies solely on the grid power as the energy source. Therefore, there is no limit on the amount of energy could be utilized, i.e., $e_{i}\left(t\right)\in \left[0,\infty \right]$. For this group, we consider the state vector dimension as $d_{1}=2$.GS: This group of equipment is equipped with grean energy harvesting capabilities. For solar energy collection, we consider solar panels with the following parameters: $p^{\mathrm{solar}}=300\;{W/m}^{2}$, $\epsilon_{0}=3.8\;\mathrm{cm}\times 9\;\mathrm{cm}$, $\epsilon_{1}=50\%$, and $\epsilon_{2}\in \left[0.5,1.5\right]$. The collected energy is stored in batteries with maximum capacity $E_{\max}=10\;J$. The dimension of the state vector $d_{2}=3$ is considered in this group.MS: The equipment in this group relies on both the grid energy and the green energy source. For the grid energy source, there’s no limit on the amount of energy could be utilized, i.e., $e_{k}^{B}\left(t\right)\in \left[0,\infty \right]$. For the green energy harvesting, we consider solar energy collection as in group GS. Similarly, the same solar panel parameters are considered here. Moreover, the collected energy is stored in the batteries with the maximum capacity $E_{\max}=10\;J$. We consider a state vector dimension of the $d_{3}=9$ for this group.

Moreover, for our three layered knowledge model, the values of *d* and *m* are obtained through validation experiment. In addition to the above parameters, the parameters shared by all parties pieces are listed below. We consider a bandwidth B=180kHz and noise power spectral density $N_{0}=-174\;\mathrm{dBm}/\mathrm{Hz}$. In addition, the loss of the channel is $g=128.1+37.6\log10(l_{y})$, where $l_{y}$ (in km) and the standard deviation of shadow fading is 8dB. With regard to computing energy consumption, we utilize associated values such as: $\varsigma =10^{-27}$, $\kappa_{y}=40$ and $\vartheta =10^{9}$. We consider the harvested energy arrival probability σ=0.7. For each piece of equipment, we assume the average size of the arrived data packets for each group is randomly generated from the range [20,60]. Within each group, the $a_{y,t}$ follows a Gaussian distribution specifically. For all the equipment, we assume $p_{y,\max}=0.01W$ and $\Delta_{\max}=30$. The simulations in the article were conducted using a MacBook Pro with an M1 chip. The code was executed on Matlab R2024a, and the MacOS system version used was Ventura 13.2.1.

### 4.2. Results and Analysis

For comparison purposes, two benchmark algorithms are compared with our proposed algorithm. The first is a Random strategy, which employs a randomly initialized policy and regular Policy Gradient (PG) updating method. To be specific, any PG methods capable of estimating policy gradient can be utilized, such as REINFORCE [23] and Natural Actor Critic (NAC) [24]. In our simulation, we adopt the NAC method as the base learning method. Additionally, we compare our proposed algorithm with the policy gradient efficient lifelong learning (PGELLA) algorithm [25]. PGELLA facilitates learning and knowledge sharing within each individual group, which is different from our cross-domain approach.

In Figure 4, we examine the initial performance improvement of the two algorithms, which we refer to as the warm start policy, compared to the random initial policy. Figure 4 shows the improvement of warm start policies of our proposed method and PGELLA over random initial policies. Both methods surpass the random initial policies. To be specific, the overall averaged results show that our algorithm can achieves 51.32% warm start policy improvement over random policy while PGELLA achieves 28.91% in general. As shown in Figure 4, for the devices in group GP, the proposed algorithm can provide a slight better performance compared to PGELLA. This is because the group GP, with lower complexity, requires less knowledge from other domains. Therefore, simpler intra-cluster knowledge sharing and migration models can provide satisfactory performance compared to cross-domain methods. Furthermore, for both the group GS and group MS, the proposed algorithm obviously outperforms PGELLA. Particularly for the MS group, the algorithm proposed can achieve a performance improvement of 90.04% compared to the random initial policy, while a 23.32% performance improvement can be achieved by PGELLA. The differing performance of the algorithm proposed and PGELLA across different groups can be attributed to the varying model complexities of these groups. The proposed algorithm enables cross-domain knowledge sharing, leading to a higher degree of exploration for complex models and the ability to migrate learned knowledge to new tasks. The enhanced performance of our method is attributed to its capacity to retain a broader spectrum of knowledge, while PGELLA is limited to providing insights specific to individual groups.

In Figure 5, a comprehensive overview of the learning trajectory is depicted across 1000 iterations for three groups. The implementation of more effective warm start policies can significantly reduce the convergence time for each group. This underscores the impact of initial policy selection on the efficiency of the learning process. In Figure 5a, it is evident that for group GP, both PGELLA and the proposed algorithm demonstrate improved warm start policies compared to the random initial policy. This enhancement effectively improves the performance of the initial policy. Additionally, both methods exhibit similar convergence speeds and final convergence performance. This similarity may be attributed to the simpler Markov models in group GP. These simpler models require less demanding algorithms to achieve satisfactory performance. In other words, PGELLA suffices for simpler models, despite lacking extensive knowledge collection and migration capabilities. Figure 5b illustrates that the enhanced capacity for broader exploration facilitates attaining globally optimal solutions. In Figure 5b, for group GS, our proposed algorithm significantly enhances the performance of the warm start policy and outperforms both PGELLA and random PG algorithm in terms of convergence speed. This improvement stems from the ability of the proposed algorithm to learn cross-domain knowledge and migrate accumulated knowledge from other domains to the current one, potentially achieving global optimality or sub-optimality and breaking out of local optima. It is worth noting that PGELLA also achieves superior results by enabling knowledge sharing within group devices, which helps overcome local optimality limitations. In Figure 5c, it can be observed that both PGELLA and the proposed algorithm achieve notable improvements in warm start policies compared to the random PG algorithm. Particularly, the proposed algorithm exhibits a significant enhancement, consistent with the performance of warm start depicted in Figure 4. Additionally, although PGELLA also contributes to the improvement in convergence rate, the algorithm proposed outperforms the other two algorithms significantly in terms of convergence speed. This is attributed to our proposed algorithm’s capability in cross-domain knowledge transfer, enabling it to achieve greater performance improvements on complex models than on simpler ones. In Figure 5d, the average performance of different algorithms across three groups is presented. Based on the averaged results, it can be observed that overall, both in terms of warm start policies performance and convergence rate, our proposed algorithm outperforms PGELLA. In a nutshell, while the performance in each single group can vary, our algorithm shows better overall performance in respect to warm start policy and convergence speed.

Figure 6 considers the default groups and environmental settings as specified in Section 4.1. In this scenario, mixed energy sources, including grid power energy and green harvested energy, are considered to minimize grid energy consumption. In a nutshell, our proposed method can achieve a better balance between grid energy consumption, battery energy consumption and AoI. By contrast, the other methods, such as PGELLA and Random methods, fail to optimize the overall performance. This proves the sustainability improvement of our proposed method. To be specific, Figure 6 shows the performance attributes for the MS group, encompassing average AoI, queue length, grid energy consumption and battery energy consumption. This group is selected due to its model complexity, allowing for a more comprehensive comparison of the algorithms. Three algorithms are considered: random PG with a random initial policy, PGELLA with intra-group knowledge sharing capability, and the proposed algorithm, the cross-domain knowledge migration algorithm. Figure 6a demonstrates the significant impact of the proposed algorithm on reducing AoI as the number of learning steps increases. In contrast, algorithms with random initial policies or PGELLA exhibit inferior performance in this regard, albeit the latter showing initial superiority compared to random initial policy. Figure 6b reveals a similar trend in average queue length reduction across all the devices in the group MS with the algorithms proposed, indicating enhanced packet processing efficiency. Notably, queue length is closely correlated with AoI. A notable difference in grid energy consumption among the three algorithms is evident in Figure 6c. While both the random initial policy and PGELLA show a decrease in grid energy consumption with increasing learning steps, our proposed algorithms exhibits an increase. However, it’s essential to highlight that this aligns with our goal of minimizing balanced cost. Our method achieves a better balance between AoI and grid energy consumption. Figure 6d indicates relatively consistent performance among the three algorithms in terms of battery energy consumption. The stability in battery energy consumption observed with the proposed algorithms is attributed to the significant performance enhancement with minimal energy consumption in the grid power network. Conversely, the battery energy consumption of the other two algorithms decreases with increasing learning steps, through with limited performance enhancement. Additionally, the PGELLA exhibits lower battery energy consumption than the random initial strategy.

As depicted in Figure 7, the performance of group GS is evaluated in terms of average AoI and average battery energy consumption. From Figure 7a, it’s evident that the performance of all three algorithms improve as the number of learning steps increases. Particularly, the initial AoI of the random PG is lower than the other two algorithms, and this trend persists as the number of learning steps increases. In Figure 7b, the battery energy consumption of all three algorithms decreases with the increase in the number of learning steps. This reduction in energy consumption is accompanied by an increase in average AoI, indicating the optimization of multiple parameters rather than a single objective. Notably, the algorithms proposed demonstrate superior optimization for average energy consumption compared to the other two algorithms. While all algorithms exhibit a significant decrease in average energy consumption with increasing learning steps, the proposed algorithms perform the best in terms of overall performance improvement across multiple parameters, as demonstrated in Figure 5b.

### 4.3. Influence of the Number of Groups

As depicted in Figure 8, the impact of the number of groups on the proposed algorithm is illustrated. It can be observed that as the number of device groups increases, the proposed algorithm offers improved initial policies. This enhancement is attributed to the increased richness of knowledge in the knowledge base with each type of group, resulting in a more diverse global knowledge base. Furthermore, since the group knowledge base relies on the existence of the global knowledge base, optimizing the global knowledge base further enhances the performance of each group. Specifically, the warm start policy performance of the proposed algorithm improves from 29.73% for a single group to 51.32% for three groups compared to a random initial policy. Similarly, as the number of groups increases, the PGELLA algorithm maintains relatively stable performance, with the warm start policy performance ranging from 28.28% for one group to 29.91% for three groups. This is because, PGELLA, as an intra-cluster knowledge learning algorithm, does not perform knowledge sharing among groups, and its warm start policy performance variation is due to the differentiated performance of different groups. Nevertheless, our proposed algorithm still achieves a better warm start policy improvement than PGELLA, demonstrating its effectiveness in handling differentiated cluster data. Furthermore, when the number of groups is insufficient, the cross-domain knowledge sharing framework has less knowledge to abstract and share, leading to a degradation in performance compared to scenarios with a larger number of groups. As a result, as the number of groups increases, the advantages of the three-layer knowledge base framework proposed become apparent, significantly outperforming the performance of PGELLA. This highlights the advantages of the three-layer knowledge base framework in handling cross-domain knowledge transfer. Therefore, the proposed algorithm exhibits more potential application scenarios and advantages when the number of groups is high.

As depicted in Table 1, the table provides a comparison of the running time and the running time difference between two algorithms: PGELLA and our proposed algorithm, for varying numbers of groups. From the table, it’s evident that as the number of groups increases, the running time of both algorithms also increases approximately linearly. With PGELLA, each additional group necessitates a complete repetition of the algorithm’s process. On the other hand, our proposed algorithm requires visiting a higher number of devices with each additional group, resulting in a longer time to visit all devices compared to PGELLA. Hence, with a single group, the runtime of our proposed algorithm (3.9225 s) is less than that of PGELLA. Yet, as the number of groups rises, PGELLA’s runtime progressively diminishes compared to our proposed algorithm. In particular, when there are three groups, PGELLA’s runtime surpasses that of the proposed algorithm by 0.5931 s. Combining the findings from Figure 8, we observe that our proposed algorithm achieves approximately a 15% performance enhancement, with a mere 4.94% increase in runtime. This suggests a perfect balance between runtime efficiency and performance improvement.

## 5. Conclusions

The article introduces a lightweight cross-domain knowledge sharing model leveraging diverse energy supply methods. It employs a three-layered knowledge base, incorporating global, group-specific, and individual policy vectors. By integrating grid, harvested, and mixed energy sources, significant improvements in warm start policy performance are demonstrated compared to random initial policies. Moreover, the collaborative nature of the global knowledge base contributes to enhanced sustainability, surpassing that of two-layered models. Considering the significant energy savings and AoI optimization achieved, our approach can facilitate the sustainability of IIoT and Industrial 4.0 initiatives. To advance in the field of cross-domain knowledge sharing, several potential research directions can be considered. These include investigating the impact of mobility, addressing privacy and security concerns, and exploring the integration of edge computing.

## Figures and Tables

**Figure 1 sensors-24-03265-f001:**
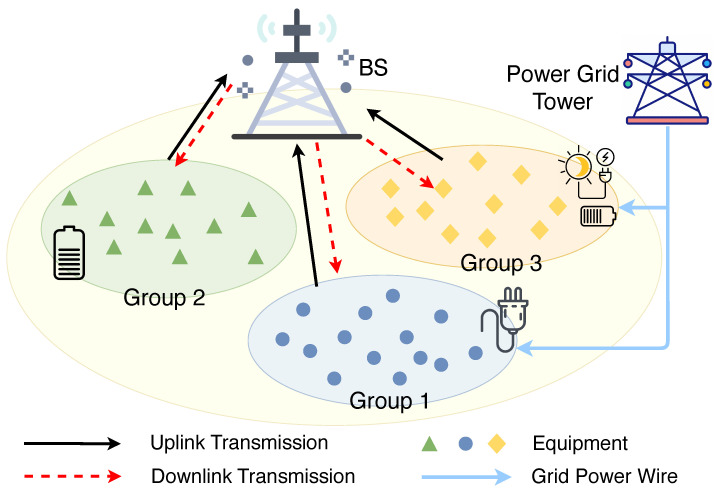
System model of three groups of smart factory equipment with diverse energy sources. Group 1 supplied by grid power wire, group 2 supplied by battery energy and group 3 supplied by grid power and green power source.

**Figure 2 sensors-24-03265-f002:**
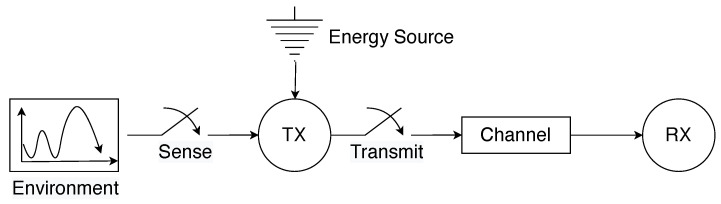
Illustrative figure of the system model representing an smart factory.

**Figure 3 sensors-24-03265-f003:**
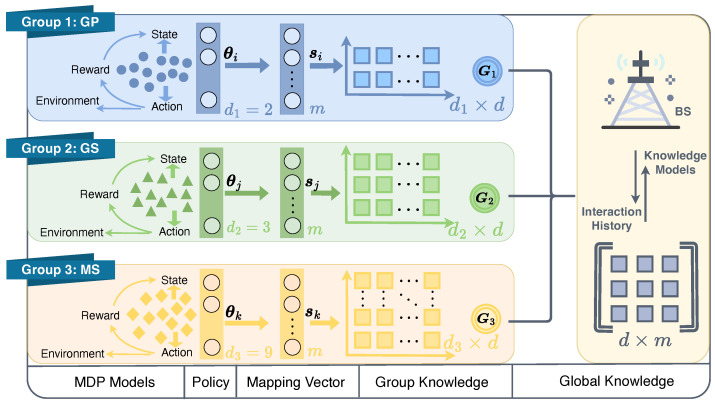
Illustration of three layers knowledge framework with varied state and/or action spaces.

**Figure 4 sensors-24-03265-f004:**
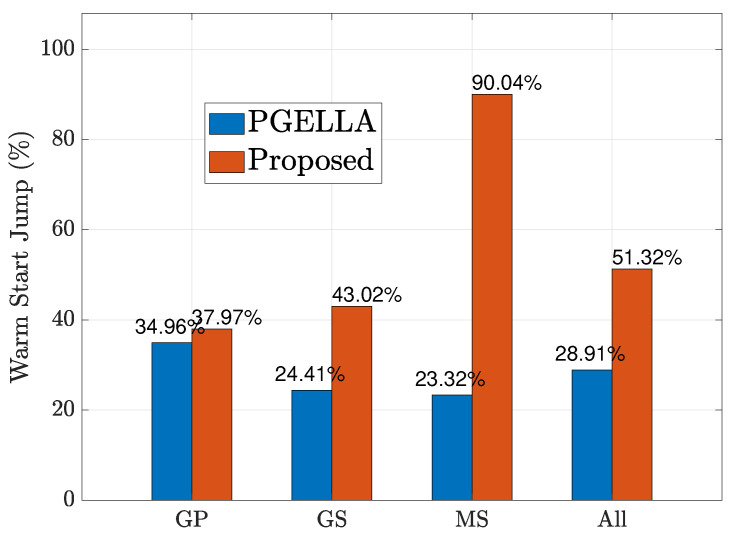
The warm start policy improvement for all the groups.

**Figure 5 sensors-24-03265-f005:**
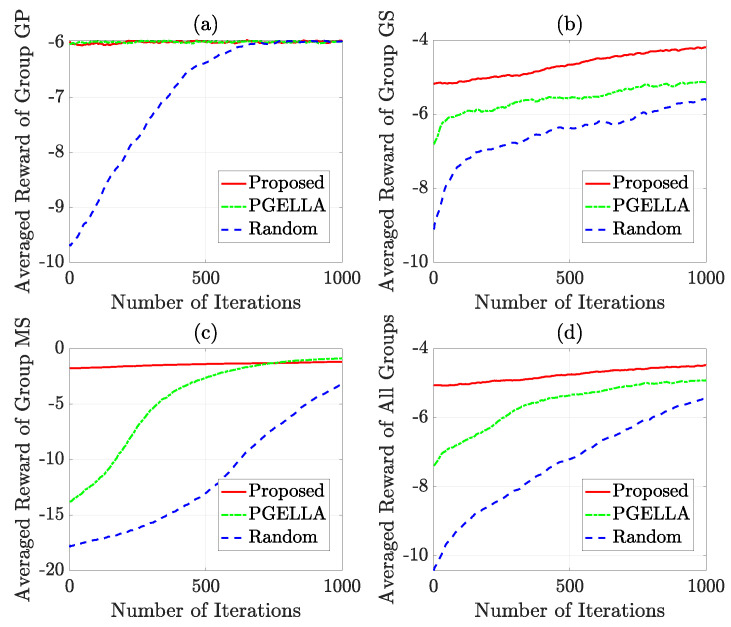
The learning methods comparison over 1000 iterations: (**a**) Group GP. (**b**) Group GS. (**c**) Group MS. (**d**) All groups.

**Figure 6 sensors-24-03265-f006:**
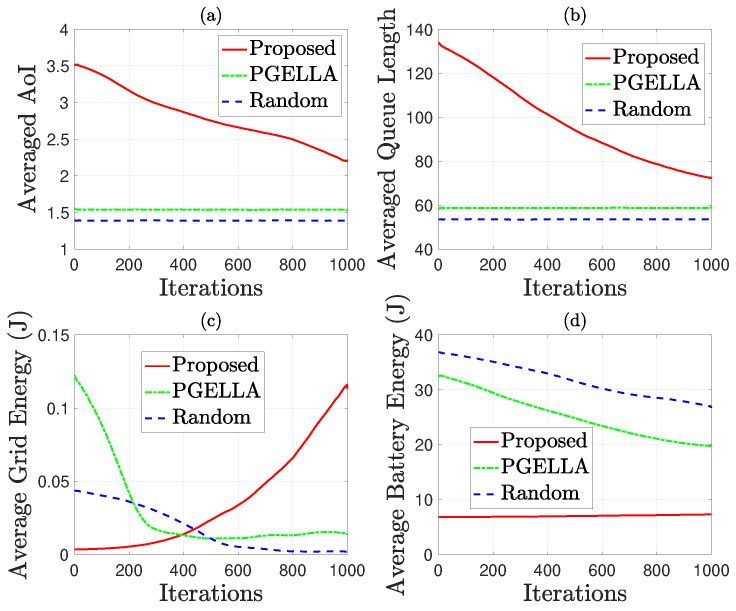
AoI and energy related performance comparison for group MS: (**a**) Average AoI. (**b**) Average queue length. (**c**) Average grid energy consumption. (**d**) Average battery energy consumption.

**Figure 7 sensors-24-03265-f007:**
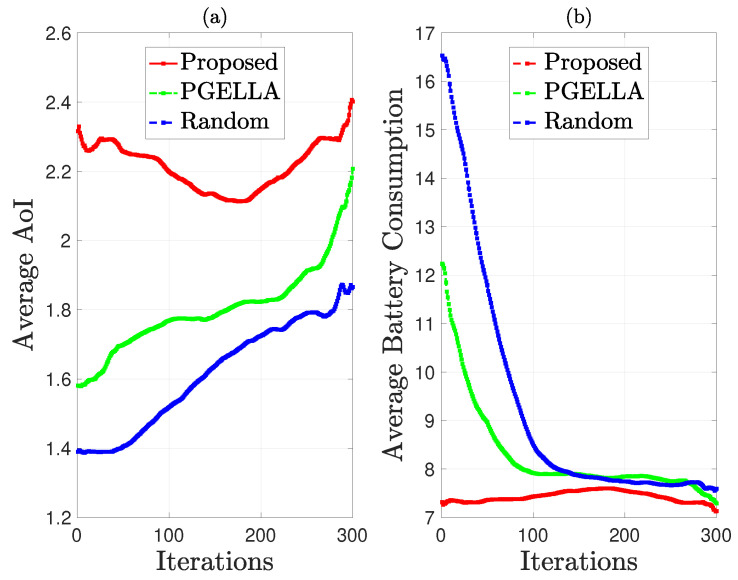
AoI and energy consumption comparison for group GS: (**a**) Average AoI. (**b**) Average total battery energy consumption.

**Figure 8 sensors-24-03265-f008:**
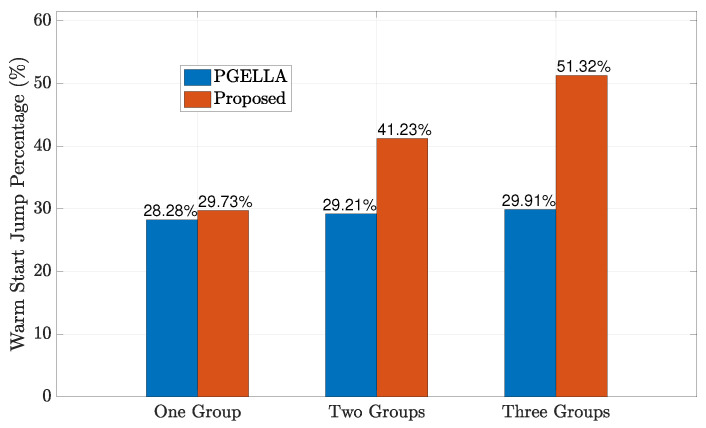
Warm start policy improvement when the number of groups increases.

**Table 1 sensors-24-03265-t001:** The Running Time Comparison.

Running Time (Seconds)	One Group	Two Groups	Three Groups
CD	3.9225	9.8620	12.5984
PG	4.0014	9.4575	12.0053
Gap	−0.0793	0.4044	0.5931

## Data Availability

Data are contained within the article.

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
