# Peer review of "Cross-Domain Knowledge Transfer for Sustainable Heterogeneous Industrial Internet-of-Things Networks"

_sensors, 2024, doi:10.3390/s24113265_

Round 1
Reviewer 1 Report
Comments and Suggestions for Authors
The research paper introduces a novel method for cross-domain knowledge transfer aimed at optimizing the trade-off between energy consumption and information freshness in smart factories using heterogeneous energy sources. Unlike traditional algorithms that rely on consistency in group scenarios, this approach allows for knowledge transfer across different states and/or actions without the need for neural spaces, making it more adaptable and less resource-intensive. The paper highlights a significant improvement in system sustainability by demonstrating a 33% increase in initial reward over random policies through simulations. This method's scalability and efficiency could potentially lead to broader applications in self-sustaining wireless networks, particularly in enhancing the operational sustainability of IoT devices across varied industrial environments.
The proposal is interesting, here are my comments:
1. In the introduction, the authors mention the utilization of three distinct groups of use cases that utilize different energy sources. Could you provide a more detailed breakdown of the experimental setups for each of these groups? Specifically, how do the differences in energy sources affect the implementation and results of your proposed cross-domain knowledge transfer method?
2. Regarding the simulation results presented in Section 4, it is noted that a 33% increase in initial reward was observed. However, the specifics of the simulation settings and the baseline comparisons are not extensively detailed. Could you elaborate on the control algorithms or previous studies against which your proposed method was benchmarked?
3. The authors could complement the search for related works related to industry 4 and 5.0 and maintenance in wireless networks, references such as:
- Yu, A., Yang, Q., Dou, L., & Cheriet, M. (2021). Federated imitation learning: A cross-domain knowledge sharing framework for traffic scheduling in 6G ubiquitous IoT. IEEE Network, 35(5), 136-142.
- Cortés-Leal, A., Cárdenas, C., & Del-Valle-Soto, C. (2022). Maintenance 5.0: Towards a Worker-in-the-Loop Framework for Resilient Smart Manufacturing. Applied Sciences, 12(22), 11330.
4. The authors describe a lightweight knowledge transfer approach that does not require neural networks, which is a significant departure from typical methods in the field. Can you provide further details on how this knowledge transfer is structured and implemented, especially how it handles the variances in state and/or action spaces across different groups?
5. In the system models section, you mention that your method enables knowledge transfer across varying state and/or action spaces. This concept is intriguing but not fully explained. Could you clarify how the model addresses and compensations for these variations to ensure effective knowledge transfer?
6. The paper lacks a clear explanation on the long-term environmental impacts and sustainability considerations of the implemented technologies, which are critical for claims of enhancing network sustainability. Could more comprehensive data or case studies be included to substantiate these claims?
Reviewer 2 Report
Comments and Suggestions for Authors
The manuscript covers an interesting R&D topic and fits the scope of the Journal. Nonetheless, the paper requires extra efforts to improve its quality and presentation. A set of comments are expounded hereafter.
“Industrial IoT” could be added as keyword.
The abbreviation IoT is not decomposed within the text; it is directly used in line 60.
An important aspect is now commented. According to the title of the manuscript, the application scenario is Industrial Internet of Things. However, in the body of the manuscript, there is no mention to Industrial IoT. In fact, IoT in itself is scarcely commented. The paradigm of Industry 4.0 is relevant and is briefly commented in the beginning of the introductory section. A manner of solving this issue is including an explicit explanation about the IIoT when talking about Industry 4.0. As it is evident, in the rest of the manuscript it should also be mentioned. Namely, it must be also commented in the Conclusions section as well as to relate the achieved results with such scenario.
Moreover, it is also suggested to support the new comments with recent publications which also highlight the interplay of Industrial IoT and Industry 4.0. For instance, the following ones, if the authors agree with the suggestion:
- RF Energy Harvesting Techniques for Battery-Less Wireless Sensing, Industry 4.0, and Internet of Things: A Review. IEEE Access 2024. doi: 10.1109/JSEN.2024.3352402
- Review of Industry 4.0 from the Perspective of Automation and Supervision Systems: Definitions, Architectures and Recent Trends. Electronics 2024, https://doi.org/10.3390/electronics13040782
- Industrial Internet of Things and its Applications in Industry 4.0: State of The Art. Computer Communications 2021. https://doi.org/10.1016/j.comcom.2020.11.016
A common practice in scientific papers consists on including a very brief description of the organization of the rest of the paper in the last paragraph of the Introduction. This contributes to the readability of the document and is strongly suggested to add it.
The environment (software, computer features) used to conduct the simulations should be briefly described in the fourth section.
A section or subsection to discuss the simulation results and their implications would be interesting.
The Conclusions section is too short. This section is not mandatory but, if included, it is recommended to enlarge it. For instance, the main limitations of the work as well as future research guidelines should be indicated..
The title of a sixth section, Templates, should be removed.
Round 2
Reviewer 1 Report
Comments and Suggestions for Authors
The authors have satisfactorily taken my comments into account.
Reviewer 2 Report
Comments and Suggestions for Authors
The new version of the manuscript has properly addressed the reviewer concerns.